# Occupational Transmission of Measles Despite COVID-19 Precautions

**DOI:** 10.3390/pathogens14060519

**Published:** 2025-05-23

**Authors:** Gabriella De Carli, Emanuela Giombini, Alberto Colosi, Maria Concetta Fusco, Eleonora Lalle, Giulia Berno, Martina Rueca, Lavinia Fabeni, Licia Bordi, Fabrizio Maggi, Maurizio D’Amato, Valentina Vantaggio, Paola Scognamiglio, Francesco Vairo

**Affiliations:** 1Regional Service for Surveillance and Control of Infectious Diseases (SERESMI)—Lazio Region, National Institute for Infectious Diseases “Lazzaro Spallanzani” IRCCS, 00149 Rome, Italy; mariaconcetta.fusco@inmi.it (M.C.F.); maurizio.damato@inmi.it (M.D.); valentina.vantaggio@inmi.it (V.V.); paola.scognamiglio@inmi.it (P.S.); francesco.vairo@inmi.it (F.V.); 2Laboratory of Virology, National Institute for Infectious Diseases “Lazzaro Spallanzani” IRCCS, 00149 Rome, Italy; dr.emanuela.giombini@gmail.com (E.G.); eleonora.lalle@inmi.it (E.L.); giulia.berno@inmi.it (G.B.); lavinia.fabeni@inmi.it (L.F.); licia.bordi@inmi.it (L.B.); fabrizio.maggi@inmi.it (F.M.); 3Department of Prevention, Local Health Authority Roma 2, 00155 Rome, Italy; alberto.colosi@aslroma2.it; 4Directorate for Health and Social Policy, Lazio Region, 00145 Rome, Italy

**Keywords:** measles, nosocomial transmission, healthcare workers, SARS-CoV-2, molecular epidemiology, vaccination, airborne pathogens, airborne transmission

## Abstract

To determine whether patient-to-doctor transmission of measles occurred in an emergency department (ED) despite isolation precautions and full personal protective equipment (PPE) during the COVID-19 pandemic, an epidemiological and molecular investigation was carried out following the identification of two subsequent cases. The N fragment was used to identify the closest whole measles genome present in the GenBank nr database and, subsequently, this was used as a reference for the reconstruction of the entire MeV sequence in the two cases studied. Seven measles-susceptible healthcare workers were on duty the day of admission of the patient, wearing full PPE. The infected doctor neither visited the patient nor entered the isolation room. The patient wore a facial respirator. No breaches in infection control procedures, or other cases among contacts, patients and healthcare workers were identified. Molecular analysis provided evidence of patient-to-worker transmission: the two B3 genome sequences showed only one mutation and no sequences of other countries were identified as phylogenetically related. Isolation precautions and full PPE were widely implemented in the ED during the COVID-19 pandemic; however, this did not prevent nosocomial transmission of measles. Vaccination of healthcare workers and enhanced ventilation should complement other preventive measures to protect workers and patients.

## 1. Introduction

During the 2017 measles epidemic, Lazio was the most affected region in Italy, with an incidence rate of 32.8 cases/100,000 inhabitants, in total 1982 cases. Among these, 129 (6.5%) were healthcare workers (HCW), representing 38.6% of national cases in the professional category: 90.9% were not vaccinated and 53.5% presented at least one complication [1]. Following the epidemic, vaccination programs were encouraged by the region [2,3,4,5], and the initiatives were widespread: hospitals and other healthcare facilities actively offered screening and vaccination with two doses, one month apart, against measles, mumps and rubella (MMR).

Regarding measles virus (MeV) genotype, before the COVID-19 pandemic, the Italian national measles and rubella surveillance system identified the co-circulation of the genotypes B3 and D8 [6].

Following the COVID-19 pandemic, a sharp decline in measles was observed in the Lazio region in 2020, with 11 cases, and in 2021 no cases were identified. In 2022, only one isolated case was detected in May, the origin of which could not be identified: IgM were confirmed but no sample was available for genotyping. The patient remained at home, and did not cause secondary cases. In October two subsequent cases were observed, one of which was an HCW working in the emergency department (ED) where the first case had been admitted. An epidemiological and molecular investigation was carried out to confirm transmission despite COVID-19 precautions.

### Description of Cases

In October 2022, a person in their thirties, confused and dehydrated, was admitted to an ED in Rome due to high fever and skin rashes. COVID-19 precautions were in place, so HCW wore respiratory protection (FFP2) and full personal protective equipment (PPE); an FFP2 mask was provided to the patient upon entry. Triage occurred in a single room, and isolation precautions were implemented. Measles was readily suspected, as the patient was not vaccinated, and serology was performed shortly after admission, showing positive IgM with negative IgG. The patient acknowledged contact with a probable case in another region, which was traced and eventually confirmed. After approximately 12 h, the patient was transferred to the National Institute for Infectious Diseases “L. Spallanzani” IRCCS (INMI), where the Regional Reference Laboratory (RRL) for Measles and Rubella Surveillance is also located, with airborne precautions, and placed in a negative pressure room. Measles was confirmed by the INMI RRL: RT-PCR was positive in whole blood and urine samples. The patient recovered after one week and was discharged without any complications.

Seventeen days later, an emergency physician working in the same hospital presented to the same emergency room reporting a 6-day high fever, resistant to antipyretics, and was immediately isolated in a single room with an FFP2 mask. During examination, a maculopapular rash was observed, along with hyperemic pharyngitis, and scarlet fever was suspected. Several serologies were performed and measles IgM was positive with negative IgG; the doctor reported no vaccination. Samples were sent to the INMI RRL and measles was confirmed; serology showed higher IgM titer and negative IgG, and urine RT-PCR was positive.

No cases of measles were identified following epidemiological investigation and active surveillance among the doctor’s contacts and residents in the area. ED admissions in the last 21 days prior to the onset of symptoms of the second case were reviewed and, apart from the first confirmed case, no further suspected cases of measles were identified; this also excluded further cases secondary to the two already identified. No other cases were identified in the population served by the hospital during the epidemiological investigation or in the entire region throughout 2022.

The doctor was on duty the day of admission of the first case, but did not know that a measles case had been seen in the ED, and had not personally visited the patient or entered the isolation room; high compliance with PPE use and infection control procedures was reported during work. Of note, thorough training on FFP2 use had been performed in the ED involving all healthcare personnel. The public health specialists of the territorial unit went to the ED, following the “fever pathway” adopted for the two cases. After triage in a single room, when patients were provided with an FFP2 mask, both cases were placed in an isolation room—there are no negative pressure rooms available in the ED—where the doors have a push-button opening, and then into an “observation unit”, where the door has to be opened with a keyboard combination. No breaches in infections control procedures were identified. The hospital’s Occupational Medicine Unit provided the measles serological status of all ED staff, including 49 doctors and 111 nurses/healthcare assistants: 4 (8.2%) and 16 (14.4%), respectively, had not been tested. Among tested HCW, one doctor (the case) (2.2%), and seven nurses/healthcare assistants (7.4%) were susceptible. Among these 8 susceptible HCW, 7 were on duty the day of the hospitalization of the first case, plus 10 (2 doctors and 8 nurses/healthcare assistants) out of the 20 who had not been tested; apart from the doctor, however, no one else contracted measles.

To verify whether nosocomial transmission had occurred, the epidemiological investigation was combined with molecular investigation by measles virus genotyping [7].

## 2. Materials and Methods

### 2.1. Viral RNA Detection and Sequencing

Measles diagnosis was performed using a commercial real-time RT-PCR (Measles Virus Real Time RT-PCR LiferiverTM, San Diego, CA, USA) according to the manufacturer’s recommendations. For sequence determination, a 450-nucleotide fragment of the N gene was amplified, using a nested PCR as previously described [8]. Genotyping was determined based on Sanger sequencing, and both INMI cases were B3.

Whole genome sequencing (WGS) was performed on residual pellet urine samples from the INMI two cases.

The full genome of MeV was reverse transcribed and amplified using OneStep RT-PCR kit (Qiagen, Hilden, Germany) by a multiplexed PCR amplicon approach, using two pools of primers specific for B3, D4 and D8 genotypes (totaling 20 overlapping amplicons with a medium length of 1000 bps) according to Ponedos et al. [9].

Libraries were then prepared starting from 10–100 ng of DNA using Ion Xpress Plus Fragment Library Kit (ThermoFisher Scientific, Waltham, MA, USA) according to the manufacturer’s instructions, and sequencing was performed on Gene Studio S5 Prime Sequencer (Thermofisher scientific Waltham, MA, USA) to obtain approximately 1 million reads per sample.

### 2.2. Data Analysis

Raw data were analyzed and all reads with average quality Phred score <20 were trimmed using Trimmomatic software v.0.39 [10]. The entire MeV genome was reconstructed by mapping reads to the reference (MN630023.1) using BWA [11]. In the subsequent analysis, only positions with a minimum coverage of 50 were considered.

The reference was selected using Blast (v. BLAST+2.16.0) on nr databases and the complete sequence closest to the Sanger fragment was selected. The assembled genomes were also manually checked using Geneious Prime v.2019.2.3., and the sequences alignment was represented as a logo using the WebLogo (10.1101/gr.849004) web application.

### 2.3. Phylogenetic Tree and Genetic Distance

Phylogenetic analysis was performed using all representative full-length sequenced genomes in B3 genotypes on NCBI (86 sequences). Furthermore, the recent complete genomes of the D8 strain sequenced in Italy were added and used as a phylogenetic outgroup. All 89 sequences were aligned with MAFFT v7.271 [12] and checked manually using the Geneious prime program. Regions not covered in the INMI genomes were excluded in all genomes. Finally, only non-redundant sequences, identified with cd-hits, were considered in the subsequent analysis. Maximum likelihood (ML) phylogenetic analysis was performed with IQ-TREE v.1.6.12 [13] and the best tree model was selected using the best DNA model included in MEGA X. The tree was built with the NT93+G model and 5000 bootstrap repetitions.

The average intra- and intergroup genetic distance of the B3 genomes detected in Europe, World, Italy and in the two INMI cases was calculated using MEGA X.

## 3. Results

### 3.1. Molecular Epidemiology

After identification by RT-PCR, a 450 bp amplicon on the N gene was sequenced with Sanger technology. The region covered by the sequencing shows that the two genomes belong to the B3 genotype and that the two fragments were identical (Figure 1A).

To validate the correlation between the two infections, the entire MeV genome of the two patients was sequenced. The N fragment, obtained by the Sanger method, was used to identify the closest whole measles genome present in the GenBank nr database (MN630023.1, Figure 1B) and, subsequently, this was used as a reference for the reconstruction of the entire MeV sequence in the patients studied.

More than 1 million MeV reads were obtained for each sample, and the entire genome was reconstructed by mapping the reads to the reference. Full coverage of 97.9% and 96.6% was achieved for Pt1 and Pt2, respectively, with an average deep coverage of 14,369.0 ± 16,678.3 and 12,076.11 ± 13,761.5 (Figure 2).

The two sequences obtained from INMI patients were found to be identical, with the exception of a mutation present in the region encoding the phosphoprotein (P) at position 2587 (covered more than 8000X), where Pt1 has C instead of T as in Pt2.

In both genomes, a region in the intergenic sequence between the matrix (M) and fusion (F) proteins showed coverage of less than 50 reads (325 and 57 uncovered nucleotides for the Pt1 and Pt2 sequences, respectively).

### 3.2. Phylogenetic and Distance Analysis

A phylogenetic tree was obtained using 87 whole MeV genomes (86 B3 genotypes and 1 D8 outgroup). Redundant genomes were eliminated, and the tree was constructed with 64 B3 genomes and 1 D8 genome. As shown in Figure 3, INMI sequences clustered separately from other sequences with a significant bootstrap of 100%.

The sequences were then divided, with respect to the country of origin, into Italian, European and non-European, and compared with the INMI sequenced cases.

The average intragroup genetic distance (calculated using the p-distance model) showed that the distance between INMI sequences was significantly smaller than the distance between sequences present in other groups (Table 1 (Panel A)). Furthermore, the average distance between groups revealed that INMI sequences were closer to European sequences than to Italian ones (Table 1 (Panel B)).

## 4. Discussion

Molecular analysis provided evidence of nosocomial, patient-to-HCW transmission: the almost complete genome sequences obtained from the samples of the two cases studied here showed only one mutation and no sequences of other countries were identified as phylogenetically related. A minimum coverage of 50 reads was achieved across most of the entire genome, and a reliable consensus was reconstructed in 97% of MeV sequences. Only one region in the intergenic sequence between M and F genes (325 and 57 uncovered nucleotides for the Pt1 e Pt2 sequences, respectively) was not covered in both sequences, near position 4900 of the reference genome. Although the diversity in intergenic regions in the wild-type strain is relatively low and these are conserved in several members of the paramyxoviruses, a high tolerance for mutagenic events was observed in the untranslated region between the M and F genes [14] consistently with the low amplification of this region that we observed. However, the two regions that the WHO identified as important for assessing epidemiological linkage were covered and the molecular results generated can be considered robust. Furthermore, in the phylogenetic tree the two INMI sequences clustered together with a 100% bootstrap, and were closer to the European and non-European sequences than to the Italian ones. This could mean that the virus came from foreign countries, or that not all the Italian circulating strains were sequenced and shared in the databases. However, the similarity was so close that hospital transmission could be confirmed by molecular and epidemiological data.

In Lazio, the adoption of strict isolation precautions against SARS-CoV-2 in ED was issued for the first time with a regional note dated 14 February 2020, imposing the use of standard precautions including respiratory hygiene, airborne, contact and droplet precautions [15], and further strengthened several times, the last of which on 31 October 2022 [16]. Thorough training was performed on proper use of PPE and facial respirators, and regularly repeated during the COVID-19 pandemic, in all hospitals, and certainly in the involved ED, by the infection control service staff.

Although all HCW wore full protective clothing and the source patient was provided with a face respirator, which has been shown to be more effective in reducing the viral spread of SARS-CoV-2 in the environment [17], the only susceptible doctor on duty contracted measles in spite of wearing full PPE, without having had any contact with the patient. Nosocomial transmission could have occurred either through the air, even after the patient’s discharge from hospital, given the long stay in the ED and the possible non-optimal adherence to the use of FFP2 due to the altered patient conditions. Alternatively, transmission could have occurred through indirect contact: the staff who treated the first case could have touched contaminated surfaces in close proximity to the patient, and subsequently opened doors with contaminated gloves; the ED doctor may subsequently have touched the contaminated buttons and inadvertently touched the mucous membranes of the face before removing the gloves. Though no evident breaches in infection control precautions could be identified, it was recommended to replace the button openings with a photoelectric opening, as well as vaccinating the remaining susceptible healthcare personnel. It is also possible that measles may have been transmitted from an asymptomatic person, although this is extremely rare and there are studies indicating no evidence that people with inapparent measles virus infections shed measles virus [18]. Furthermore, this circumstance should have been highlighted by the occurrence of other cases in hospital or in the community. Active surveillance was strengthened by alerting general practitioners, pediatricians, and hospitals for the timely identification of suspected cases, and no further cases have been identified in the Lazio region throughout 2022.

This outbreak confirms the absolute importance of primary prevention based on vaccinating susceptible HCW against measles and not relying solely on the effectiveness of personal protective equipment and infection control precautions. Indeed, in the case of SARS-CoV-2 even N95 masks were not able to completely block the transmission of virus droplets/aerosols even when completely sealed [17]. If any additional preventive measure could be suggested in the ED against pathogens that transmit through the air [19], and particularly against measles, whose infectious emission rate ranks first among respiratory pathogens [20], this would be enhanced ventilation using engineering controls [21].

## 5. Conclusions

As it already happened with SARS [22], isolation precautions and use of full PPE including facial respirators were widely implemented in the ED during the COVID-19 pandemic; however, this did not prevent nosocomial measles transmission to occur. HCW, and particularly those who work in the ED, are the frontliners in healthcare delivery, and should therefore be vaccinated to act as a barrier against the spread of diffusive infectious diseases, such as measles, to protect themselves and their patients. Furthermore, they represent a reliable source of information on the safety and effectiveness of vaccines. Their acceptance of measles vaccination will convey a stronger message to the general population, helping to curb the epidemic and move towards measles elimination.

## Figures and Tables

**Figure 1 pathogens-14-00519-f001:**
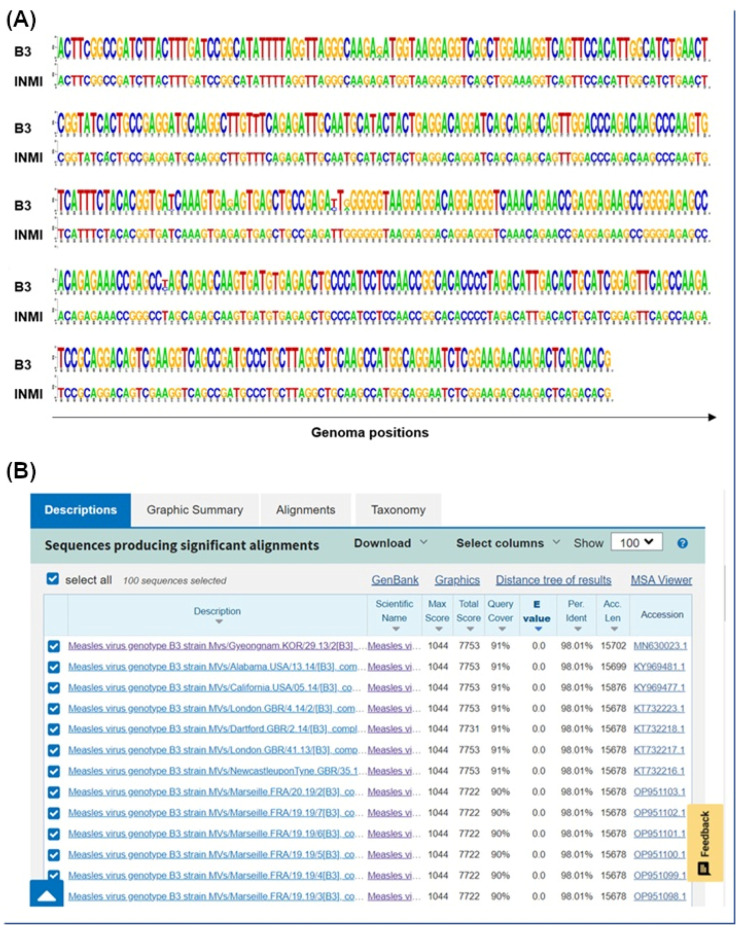
Sanger fragment molecular analysis. (**A**) Sequence logos based on the nucleotide frequencies in the two N fragments sequenced with Sanger technology (INMI) vs. 64 N regions of B3 genotype genomes. (**B**) BLAST results obtained aligning measles Sanger fragment on nr database.

**Figure 2 pathogens-14-00519-f002:**
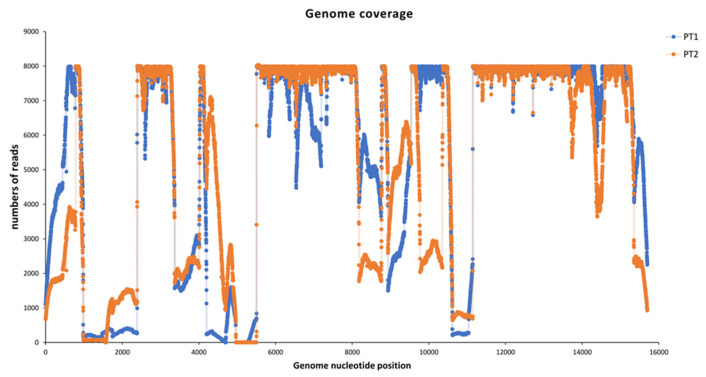
Graphic representation of the nucleotides coverage of MeV in the two patients, numbered according to the reference MN630023.1 and using a maximum threshold of 8000 reads.

**Figure 3 pathogens-14-00519-f003:**
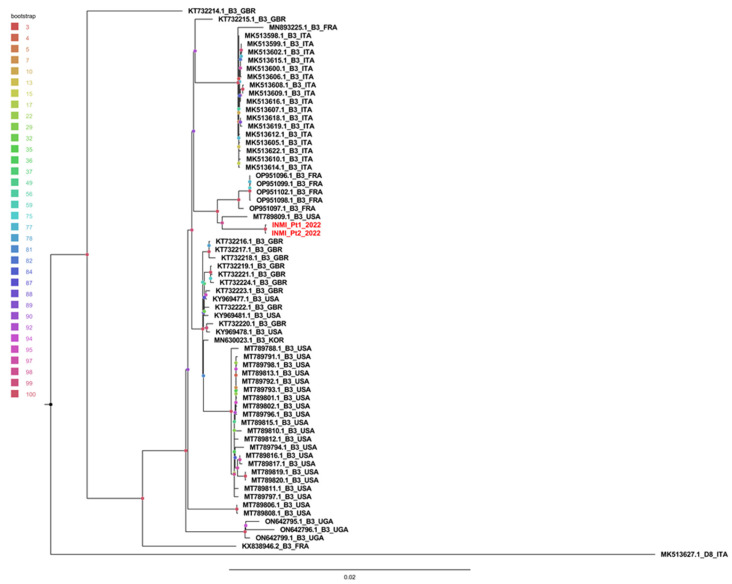
Phylogenetic analysis of the whole genome of measles virus (MeV) built with the 64 B3 genome and one D8 as outgroup. The tree was built using the bootstrap method and the node bootstrap was identified with colored dots. The color gradient is reported in the legend. The two INMI sequences are reported in red.

**Table 1 pathogens-14-00519-t001:** Mean genomic distance and standard deviations calculated intragroup (panel A) and intergroup (panel B) using the p-distance model, by grouping the sequences with regard to geographic location (Italian, European, World and INMI).

Panel A—Average intragroup genetic distance					
Geographic location		Mean genomic distance		Standard deviation	
Europe		0.0069		33.49 × 10^−5^	
Italy		0.0003		6.56 × 10^−5^	
World		0.0042		26.68 × 10^−5^	
INMI		0.0001		9.0681 × 10^−5^	
Panel B—Average intergroup genetic distance					
Geographic location	Europe		Italy		World
Europe					
Italy	0.0075 ± 0.0005				
World	0.0070 ± 0.0003		0.0078 ± 0.0006		
INMI	0.0088 ± 0.0006		0.0097 ± 0.0007		0.0095 ± 0.0007

## Data Availability

Sequence data obtained in this study have been deposited into GenBank (accession numbers: PP446317—PP446318).

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
