# Peer review of "Occupational Transmission of Measles Despite COVID-19 Precautions"

_pathogens, 2025, doi:10.3390/pathogens14060519_

Round 1

Reviewer 1 Report

Comments and Suggestions for Authors

The manuscript is well written, and the methodology appropriate and thorough. The authors also draw valid conclusions and recommendations. The latter are as would be expected, that all staff should be vaccinated and ventilation should enhanced.  There are no criticisms of the study or manuscript. Overall, this is an important demonstration of how despite all the standard precautions, that measles as one of the most infectious viruses can get through these barriers.

Author Response

The manuscript is well written, and the methodology appropriate and thorough. The authors also draw valid conclusions and recommendations. The latter are as would be expected, that all staff should be vaccinated and ventilation should enhanced.  There are no criticisms of the study or manuscript. Overall, this is an important demonstration of how despite all the standard precautions, that measles as one of the most infectious viruses can get through these barriers.

Response: we are deeply grateful to Reviewer 1 for the encouraging comments, indeed we meant to support with  specific data that standard precautions and facial respirators must be supplemented by primary prevention and possibly by engineering controls to prevent measles transmission in healthcare settings.

Reviewer 2 Report

Comments and Suggestions for Authors

This is a very good measles transmission report in HCW despite protective measures. This fact supports the consideration of this paper for publication, but there are some problems in the discussion of forms of transmission which do not comment of asymptomatic  infection of other personal contaminating the environment. This must be discussed. 

Aside to this, the huge sequence information is very good but it is difficult to find the mutation in the sequence graph, please insert arrows to help the reader. 

Author Response

Comment 1: This is a very good measles transmission report in HCW despite protective measures. This fact supports the consideration of this paper for publication, but there are some problems in the discussion of forms of transmission which do not comment of asymptomatic  infection of other personal contaminating the environment. This must be discussed.

Response 1: Thank you for pointing this out. We agree with this comment. Therefore, we have added a paragraph addressing this issue and added a reference [17] to a study on transmission from asymptomatic persons. [page 8, 2nd paragraph, lines 240-246].

Comment 2: Aside to this, the huge sequence information is very good but it is difficult to find the mutation in the sequence graph, please insert arrows to help the reader. 

Response 2: Agree, however the mutation is not in the fragment shown in the figures. We have, accordingly, added the details in the Results section [page 5, 2nd paragraph, lines 166-168].